# Tricarboxylic Acid Cycle Intermediates and Individual Ageing

**DOI:** 10.3390/biom14030260

**Published:** 2024-02-22

**Authors:** Natalia Kurhaluk

**Affiliations:** Department of Animal Physiology, Institute of Biology, Pomeranian University in Słupsk, Arciszewski St. 22 B, PL 76-200 Słupsk, Poland; natalia.kurhaluk@upsl.edu.pl

**Keywords:** tricarboxylic acid cycle intermediates, ageing mechanisms, individual physiological reactivity, bioenergetic mechanisms of ageing, individual ageing processes, anti-ageing therapy

## Abstract

Anti-ageing biology and medicine programmes are a focus of genetics, molecular biology, immunology, endocrinology, nutrition, and therapy. This paper discusses metabolic therapies aimed at prolonging longevity and/or health. Individual components of these effects are postulated to be related to the energy supply by tricarboxylic acid (TCA) cycle intermediates and free radical production processes. This article presents several theories of ageing and clinical descriptions of the top markers of ageing, which define ageing in different categories; additionally, their interactions with age-related changes and diseases related to α-ketoglutarate (AKG) and succinate SC formation and metabolism in pathological states are explained. This review describes convincingly the differences in the mitochondrial characteristics of energy metabolism in animals, with different levels (high and low) of physiological reactivity of functional systems related to the state of different regulatory systems providing oxygen-dependent processes. Much attention is given to the crucial role of AKG and SC in the energy metabolism in cells related to amino acid synthesis, epigenetic regulation, cell stemness, and differentiation, as well as metabolism associated with the development of pathological conditions and, in particular, cancer cells. Another goal was to address the issue of ageing in terms of individual characteristics related to physiological reactivity. This review also demonstrated the role of the Krebs cycle as a key component of cellular energy and ageing, which is closely associated with the development of various age-related pathologies, such as cancer, type 2 diabetes, and cardiovascular or neurodegenerative diseases where the mTOR pathway plays a key role. This article provides postulates of postischaemic phenomena in an ageing organism and demonstrates the dependence of accelerated ageing and age-related pathology on the levels of AKG and SC in studies on different species (roundworm *Caenorhabditis elegans*, *Drosophila*, mice, and humans used as models). The findings suggest that this approach may also be useful to show that Krebs cycle metabolites may be involved in age-related abnormalities of the mitochondrial metabolism and may thus induce epigenetic reprogramming that contributes to the senile phenotype and degenerative diseases. The metabolism of these compounds is particularly important when considering ageing mechanisms connected with different levels of initial physiological reactivity and able to initiate individual programmed ageing, depending on the intensity of oxygen consumption, metabolic peculiarities, and behavioural reactions.

## 1. Introduction

Ageing is the world’s No. 1 killer. It is a genetically determined long-term biological process [1]. All living things—from the simplest biological entities to such a complex entity as an organism—undergo this process. Ageing can be physiological (natural) and pathological, i.e., accelerated or premature [2]. The processes of negative senescence/negligible senescence are considered by many authors [3,4] and link these phenomena with a mortality risk that remains stable or decreases with age. These processes, as noted by some authors [5] are observed in some wild animals. It is noted that the age-independent mortality of these groups of animals may result in an abnormally long maximum lifespan. The authors [5] emphasise that these features of ageing individuals may be incompatible with generally accepted evolutionary theories of ageing. These mechanisms regarding the rate of ageing of wild animals with a low probability of recovery may be comparable for any ageing models and can be effectively used to improve and prolong human health [6]. Physiological ageing differs from premature ageing, first of all, by the fact that it is not burdened with diseases and, therefore, it does not need to be treated. It is a process that affects all cells of the human body. Anti-ageing biology and medicine programmes incorporate the latest knowledge in molecular biology, genetics, immunology, endocrinology, nutrition, therapy, cosmetology, and other specialisations [7].

Elucidation of the mechanisms of ageing and determination of metabolic therapies, such as calorie restriction [8], fasting [9], exercise [10], and ketogenic diets [11], have been convincingly shown in the literature to prolong longevity and/or health [12]. Individual components of these effects are postulated to be related to energy supply and free radical production processes. However, their convincing benefits, as shown in studies from different systematic groups [13], are still limited and their relationship to the underlying mechanisms of ageing is not completely clear.

There are so-called biomarkers of ageing that help to estimate a person’s biological age—these are measurable physiological parameters that can be used to qualitatively and quantitatively determine/measure the process of human ageing [14]. Knowing the deviation of a biomarker from the norm, it can be adjusted and thus it is possible to slow down the ageing process using data-intensive technologies [15].

The convincing differences in the mitochondrial characteristics of energy metabolism in animals with different levels (high and low) of physiological reactivity of functional systems shown in the literature are related to the state of three leading regulatory systems providing oxygen delivery to tissues: the respiratory system, the state of the cardiovascular system, and the transport function of blood [16,17]. Hypoxic damage that accompanies many pathologies is especially intensified in old age subjects. Importantly, these differences in the resistance to oxygen deficiency imply differences in the implementation of ageing programmes resulting from differences in the systemic inflammatory response, depending on species, organ, sex, age, and individual characteristics [18]. In combination, all these functional–metabolic characteristics of low-resistant animals may be the cause of such diseases as diabetes, atherosclerosis, coronary thrombosis, ketoacidosis, etc., and the possibility of their development is much higher than in animals with high resistance, which classifies the former into a risk group with rapid fatigue and reduced performance. It is important that short- and long-term bioenergetic mechanisms of adaptation to hypoxia implemented during the life of each individual differ, which in turn can intensify the ageing programmes [19,20].

Since it is difficult to overestimate the crucial role of the tricarboxylic acid (TCA) cycle (i.e., Krebs cycle, citric acid cycle) in the energy metabolism of cells, amino acid and protein synthesis, epigenetic regulation, cell stemness and differentiation, fertility and reproductive health, as well as the metabolism associated with the development of pathological conditions and, in particular, cancer cells, the aim of this review was to address the issue of ageing in terms of individual characteristics related to physiological reactivity. What role can substrates of the Krebs cycle play in these processes? This review discusses our reports and other recent data on these relationships to suggest the causes of loss of efficiency and ways to overcome it, using the example of individual physiological reactivity, which may determine the direction of individual ageing-based processes on the structure–function relationships of energy metabolism. Consideration of the relationship between these interdependent processes was the additional aim of this study.

## 2. Biological Mechanisms of the Ageing Process

There are several theories of ageing [1,21,22], but one of the most popular and most widely supported approaches has been proposed by López-Otín et al. [23], who defined ageing in different categories and explained the interactions between various factors stimulating the development of age-related changes and diseases. For example, Gonçalves et al. [24] described the major causes of ageing related to epigenetic changes associated with gene expression [25], e.g., global DNA methylation in genes [26], epigenetic alterations, genomic instability, and mitochondrial dysfunction related to the mitochondrial DNA copy number, depletion of telomeres on chromosomes or telomere attrition, and changes in proteostasis processes leading to the accumulation of cellular waste by Hsp70 and Hsp72 markers [27]. The relative contribution of energetic and functional barriers to the evolution and manifestation of ageing is discussed in the work of the authors [28]. The results of experiments on interorgan systems to determine the longevity of species in adapting to ecosystems presented in several papers highlight the role of heterochronic parabiosis, systemic factors such as DAMP, TF-like vascular proteins, and inflammation, and they focus on the ageing clock located at different levels of organisation from individual cells to the brain [29,30]. The uncontrolled free radical production and oxidative stress postulated by many authors [22] lead to mitochondrial dysfunction and, consequently, impaired energy production [31].

The ageing process is characterised by the appearance of ageing signs in organs and systems with a different onset time, a different degree of expression of age-related changes in organs or their separate parts, and a different speed. These changes can be observed in tissues and systems, such as the blood, heart, liver, and skin [32]. The processes of early ageing are distinguished by the presence of an aetiological factor, which leads to the initiation of pathological processes by the reduced efficiency of compensatory mechanisms and a more pronounced limitation of adaptation to changing environmental conditions and, finally, by more pronounced chronic and acute changes in the organism when the changes relate to one organ or system (Figure 1).

The causes of ageing are associated with the impaired nutrient sensitivity of cellular functions, inducing cellular ageing accompanied by chronic inflammation and loss of tissue regeneration. Another important process is stem cell depletion and altered intercellular communication, e.g., chronic inflammation and dysfunctional cell behaviour in the GDF-15 analysis of mitochondrial dysfunction [33], cellular senescence as a homeostatic biological process that has a key role in driving ageing [34,35,36], and altered intercellular communication [37]. Significant aspects of mitochondrial changes during ageing as elements of some anti-senescence strategies are considered in study [38]. Other important signs of ageing that are possible to add to the existing paradigm have been reported by Gonçalves et al. [24]. The first proposed new sign is impaired autophagy. The process of autophagy during ageing is a phenomenon in which cells consume their own components (organelles) as a fuel source, and it is associated with dysregulation of splicing, which builds RNA from DNA, disturbances in the microbiome [39], and dysfunction of laminin, i.e., the nuclear envelope protecting DNA.

## 3. Biomarkers of Ageing

Ageing is a series of processes that include direct DNA damage, the accumulation of cellular waste, metabolic errors, and imperfect repair and the body’s response to these processes, resulting in the development of known signs of ageing and age-related diseases. According to the Ageing Biomarker Consortium, the definition of ageing consists of three main combinations of biological parameters: assessment of age-related changes, tracking the physiological ageing process, and prediction of the transition to pathological status [40].

The clinical descriptions of the top markers of ageing, which are currently being investigated, include the following parameters [24]: DNA copy number, telomere length, global DNA methylation, Hsp70 and Hsp72, insulin-like growth factor 1 (IGF-1) [41,42], SIRT1 (Sir2) as an NAD^+^-dependent deacetylase playing critical roles in a broad range of biological events [43] and a deregulated nutrient-sensing level, GDF-15, CD4^+^, and CD8^+^ cell percentages [44] as a cellular senescence level [45,46], and circulating osteogenic progenitor (COP) cells as a marker of stem cell exhaustion. Next in this line are the levels of IL-6 [47], CRP, and TNF-alpha as evidence of altered intercellular communication [48]. The authors indicated that IGF-1, SIRT1, GDF-15, IL-6, CRP, and TNF-alpha were the most efficient biomarkers.

The thickness of the intima–media complex (IMC) of the common carotid artery characterises vascular ageing and is a marker of early atherosclerotic lesions in the vascular wall. The IMC not only reflects local changes in the carotid arteries, but also indicates the prevalence of atherosclerosis. The greater the IMC thickness, the higher the likelihood of ischaemic stroke and transient ischaemic attack [49]. Such parameters as blood pressure, body mass index, and waist circumference are often associated with biomarkers of ageing. In clinical practice of cardiovascular diseases, high insulin-like growth factor-1 also seems to be a biomarker of ageing; it may indicate the presence of inflammatory processes that accelerate ageing and the levels of glycosylated haemoglobin, IL-6, C-reactive protein, ferritin, and homocysteine [50].

Among the many factors of ageing, mitochondrial dysfunction, the accumulation of cells that have already lost the ability to divide (senescent cells), genome instability, deregulation of major genes (epigenetic ageing), glycation (protein cross-linking and aggregation), systemic inflammation, chronic hypoxia, and chronic stress are now being highlighted [51,52]. This ranking of the main biomarkers was added by authors who assessed 44 markers in seven categories, categorising 19 as high, 22 as moderate, and three as low [53].

A very important molecular pathway that accelerates ageing is the activation of the mTOR signalling pathway [54,55]. The mTOR pathway is activated by the consumption of large amounts of the amino acid methionine (red meat, sausages), which, on the one hand, is an essential amino acid and therefore essential for the normal functioning of the body [56]. At the same time, an excess of methionine, especially at a certain age (approximately after 40 years), provokes accelerated protein synthesis; hence, the cell grows, depending on its specialisation, multiplies and, ultimately, finds itself in a stressful situation, because this protein cannot specialise [57]. Since this is a very energy-intensive process, extremely important mechanisms for controlling “breakdowns”, such as autophagy (the process of cell self-cleaning from cellular debris), are disabled [58]. If the intake of methionine can somehow be inhibited (for example, by eating poultry instead of red meat), the processes that promote cell self-cleaning and repair of “breakdowns” are activated, stress resistance increases, and the chances of being healthy increase (Figure 2).

The systematic research on biological ageing targeted at understanding the underlying mechanisms of stressors and energy supply processes continues, especially after the publication of studies showing that chronic stress at an early age reduces life expectancy. The concept proposed by [59] explains that chronic exposure to cortisol at an early age contributes to persistent changes in the stress response system and compromises the regulation of central immune system genes whose expression controls inflammation. Since the body’s regenerative capacity is known to depend on the state of immune regulation, an in-depth understanding of the immune system responses to stressors may explain the link between chronic stress and impaired regeneration. The potential of research in studying the role of klf9 in the development of age-related neurodegenerative diseases, such as Alzheimer’s disease, has been underlined. It has been suggested that klf9 is a central gene in understanding the mechanism of the optimal regulation of inflammation and the peculiarities of changes in this typical pathological process under the influence of stress factors in the early stages of organism development. This work highlights the potential for synergy that arises between scientists who collaborate in the study of ageing and regeneration [59].

## 4. Accelerated Ageing and Age-Related Pathology

Postischaemic phenomena in an ageing organism have been convincingly demonstrated for heart tissue in comparison with skeletal muscle and other tissues (e.g., liver), which differ in functional metabolic specificity; in an ageing organism, myocardial dysfunction is the result of the development of significant metabolic disorders in the myocardium during ischaemia and reperfusion to a degree that is not yet sufficient for necrosis but significantly impedes the restoration of its functional state caused by autophagy in the heart [60]. This means that metabolic correction with drugs that directly contribute to the normalisation of an impaired myocardial metabolism and can provide a more rapid and effective normalisation of the functional state of the heart significantly reduces the risk of the severe consequences of heart disease associated with hypoxia and subsequent myocardial infarction [61]. Biology and clinical medicine today have made quite a big step towards studying the problems of ageing, and it makes sense to try to manage the error accumulation programme and the genetic programme. The first programme of the accumulation of negative tendencies is associated with hormonal imbalance, chronic inflammatory diseases, metabolic disorders, stress, and behavioural reactions such as smoking, eating, etc. [62].

The search for effective means and methods of preventing premature ageing continues. To treat and prevent premature ageing, geriatric agents or geroprotectors are used, which have a stimulating effect on the ageing organism, normalise disturbed functions of organs and systems, improve metabolism, and increase compensatory capabilities [63]. There are many substances known to have the ability to slow down the ageing process and thereby increase life expectancy. These geroprotectors include vitamins, anabolic agents, biogenic stimulants, adaptogens, hypolipidemic agents, and peptide bioregulators of the cytomedine class [64].

## 5. Krebs Cycle as the Key Component of Cellular Energy and Ageing

The tricarboxylic acid cycle (TAC), or Krebs cycle, was discovered in 1937 by Hans Krebs. The reactions of this cycle take place in the matrix of the mitochondria (Mt) and are the main source of reducing equivalents in the respiratory chain [65]. The oxidation of intermediate products of protein, fat, and carbohydrate catabolism occurs in the mitochondrial matrix. Thus, fatty acids are converted into acyl derivatives of coenzyme A, which are oxidised in the Mt matrix. The breakdown and oxidation of carbohydrates is accompanied by the formation of ATP and pyruvic acid (PIR). The conversion of PIR involves its oxidative decarboxylation in the Mt matrix and the incorporation of the resulting acetyl-CoA into the TAC. During the hydrolysis of proteins, acetyl-CoA, α-ketoglutarate, fumarate, and succinate are formed along with amino acids. The final oxidation of these compounds also occurs in a cyclic reaction system called the TAC [66].

The Krebs cycle begins with the formation of citrate from oxaloacetate and acetyl-CoA and ends with the formation of carbon dioxide, water, and oxaloacetate regeneration. The condensation of oxaloacetate from acetyl-CoA, which is formed during the oxidation of PIR, fatty acids, and amino acids, leads to the regeneration of citrate and supports the cycle [67]. Krebs cycle reactions are accompanied by the transfer of hydrogen atoms to NAD^+^ molecules. The reduced form of NAD^+^ is the main intermediate between the Krebs cycle and the respiratory chain located in the inner membrane of the Mt. Aging is associated with the development of oxidative stress, inflammation and impaired Ca2+ control in the failing heart as shown by Bhullar et al. [68].

During the transition of the cell to the activation of physiological functions, the sequence of TAC reactions changes and the initial enzymes of the cycle, such as citrate synthase and isocitrate dehydrogenase, already do not determine the general rate of this cycle, and the limiting stage in this process consists in the oxidation of one of its intermediates, i.e., succinic acid (SC). Respiratory complex II (succinate dehydrogenase (SDH), canonically SDHA-SDHB-SDHC-SDHD, but with exceptions) is a heterotetrameric membrane-spanning enzyme, as postulated by Iverson et al. [69].

As reported by a number of authors [70], the activation of SC oxidation in these conditions is determined by the possibility of rapid renewal of its pool due to the transamination of glutamic, oxalic, and acetic acids with subsequent oxidative decarboxylation of alpha-ketoglutarate (KGL) to SC. It has been shown that the oxidation of succinate by mitochondria can generate a higher protonmotive force than can the oxidation of NADH-linked substrates [71], which may result from substantial succinate oxidation in vivo in pathological conditions.

The data on the structural organisation of TAC enzymes in the matrix, which are organised into a multi-enzyme complex, i.e., a “metabolon” [72,73], are in good agreement with this hypothesis. At the same time, dicarboxylic acid oxidation enzymes and aspartate aminotransferase form a closely associated aggregate in the centre of the complex, which creates conditions for accelerating their interaction. Thus, TAC enzymes act as a source of reducing equivalents entering the respiratory chain and ensuring the formation of an electrochemical potential gradient on the inner membrane of the Mt as a universal form of energy that provides the energy and transport functions of the Mt [66].

Recent data on the maintenance of homeostasis in the regulation of metabolism and cell death, in which sestrins, maintaining intracellular homeostasis through AMPK and mTOR kinases, play a leading role, have led to a conclusion about the role of sestrins in ageing and disease protection. Sestrins, i.e., mTORC1 inhibitors and stress-responsive proteins, are involved in the control of ageing, and Sestrin2 is a member of a family of stress-responsive proteins. These proteins control cell viability, have antioxidant activity, and take part as regulators of the mammalian target of the rapamycin protein kinase (mTOR) pathway. Thus, it has been demonstrated that the inactivation of Sestrin2, which regulates redox homeostasis and apoptosis in response to various stresses, can reduce ATP production [74]. Sestrin2 causes a decrease in both oxidative phosphorylation and glycolysis [75]. It seems that two amino acids, aspartate and glutamate, which are directly produced from the TCA cycle, can activate the mTOR [76,77]. Thus, the activation of sestrins in response to stress likely plays an important role in maintaining energy production via the TAC.

## 6. mTOR, Ageing, and Metabolism

DNA damage is known to stimulate ATP production by maintaining the oxidative phosphorylation chain, which may be essential for repair processes [78], and sestrins may participate in this process, as these proteins are activated in response to DNA damage [79]. Metabolic disorders in mitochondrial malfunctions, the suppression of glycolysis, or insufficient supply of essential nutrients, such as glucose or amino acids, lead to the induction of Sesn2 caused by the activation of transcription factors ATF4 and NRF2 [80]. Genetic or pharmacological inhibition of the mTORC1 kinase has been shown to increase the lifespan in most eukaryotic organisms studied, including yeast, flatworms, flies, and mice [81], in the nuclear ageing program. Some studies have shown increased longevity in a wide range of organisms due to calorie restriction, defined as a reduction in nutrient intake [82]. Given the critical role of the mTORC1 in nutrient and insulin sensing [83], it has been speculated that the beneficial effects of calorie restriction on the lifespan are also related to reduced mTORC1 signalling through overlapping mechanisms. Important issues related to caloric restriction and the activation of autophagy, which increases longevity by delaying the onset of age-related diseases in most living organisms, are discussed in a paper [84]. For these metabolic pathways in yeast [85] and cellular senescence processes in nematode, the protein kinase CK2 has been proposed [84].

Since ageing is closely associated with the development of various age-related pathologies, such as cancer, type 2 diabetes, and cardiovascular and neurodegenerative diseases [86], the mTORC1 kinase plays a key role in the regulation of ageing in this case. Sesn2, in turn, maintains cell viability in the conditions of ischaemia as well as glucose and amino acid starvation. Although the contribution of sestrins to the defence against cellular death is in many cases related to the regulation of the AMPK-mTORC1, which leads to the inhibition of biosynthetic processes in the cell and supports catabolic processes aimed at energy production and the repair of cellular structures, it is possible that Sesn2 in glucose starvation protects against necrotic death through a mechanism by controlling mitochondrial function and maintaining mitochondrial respiration in stress conditions [75].

Can TCA metabolites be used to effectively and efficiently influence biological age? Understanding the molecular mechanisms of ageing and the role of nutrition and supplements that can slow down these processes often helps people stay healthy. The following important links between TAC metabolites and the mTOR as the main target and sestrin are related to the concepts that metabolic interventions deplete acetate stores and probably reduce the conversion of oxaloacetate to aspartate, thereby inhibiting the mammalian target of the rapamycin (mTOR) pathway and enhancing autophagy [54]. Another important finding relates to the effects of glutathione, which promotes autophagy and prevents AKG accumulation, supporting stem cell maintenance [65,87].

The previously demonstrated negative effects of SC accumulation [88,89] contribute to slowing down DNA hypermethylation, promoting the repair of DNA double-strand breaks, reducing inflammatory and hypoxic signalling, and reducing dependence on glycolysis. The following dependencies of accelerated ageing and longevity decline processes are also discussed in the literature as progressive damage to aconitase, inhibition of succinate dehydrogenase, and suppression of hypoxia-inducible factor-1α and phosphoenolpyruvate carboxykinase, as shown by Jia et al. [90]. Possibly due to these mechanisms, metabolic interventions using TAC metabolites may slow down ageing and increase longevity. Conversely, in the case of overnutrition or oxidative stress, these processes act in the opposite direction, accelerating ageing and worsening longevity. Therefore, consideration of the involvement of Krebs cycle metabolites in the basic processes of cellular energetics should be examined with reference to the specific involvement of the leading metabolites, i.e., ketoglutarate and succinate, the role of which has been convincingly demonstrated in a number of studies and is given below.

## 7. α-Ketoglutarate, Energy Metabolism, and Ageing

The key role of α-ketoglutarate in TAC activity for mitochondrial respiration has recently been shown in experiments with labelled TAC metabolites in brain mitochondria oxidising a mixture of pyruvate+glutamate+malate substrates, which caused a significant increase in α-ketoglutarate (AKG) content [91,92]. The peculiarities of TAC functionality, which operates as two conjugated cycles and oxidises overamidation substrates, were previously reported by Yudkoff et al. [93] in rat brain synaptosomes and were examined as fluxes and interactions with aspartate aminotransferase and the malate/aspartate shuttle. The authors suggested that, in the presence of glutamate+pyruvate, the cycle operates as two coupled cycles, with the first cycle starting from α-ketoglutarate to oxaloacetate and the second cycle operating from oxaloacetate to AKG. These values in the first cycle were 3–5-fold higher than the flux between oxaloacetate and 2-oxoglutarate measured in the presence of glucose. Thus, during oxidation of the glutamate+pyruvate substrate mixture, activation of the α-ketoglutarate dehydrogenase complex (KGDHC) and succinate dehydrogenase (SDH) can significantly increase the rates of metabolite flux through the TCA and respiratory chain during oxidative phosphorylation.

The specific functioning of the substrate–enzyme complexes of the Krebs cycle, as well as the activity of SDH in the TCA, can be inhibited by endogenous oxaloacetate (OAA) in a process named intrinsic inhibition of SDH [94], which limits the rate of the entire TCA in metabolic states 3 (phosphorylating respiration) and 3P (uncoupled respiration). The ability of glutamate and pyruvate together and singly to overcome SDH inhibition has been linked to the metabolic removal of OAA in citrate synthase and aspartate and alanine aminotransferase reactions.

The second limiting point in the TCA is the reaction catalysed by KGDHC, as shown by Sheu and Blass [72]. It has previously been shown that the activity of KGDHC is the lowest of all the TCA enzymes [95] and is controlled by the presence of α-ketoglutarate and its affinity for KGDHC, which is controlled by Ca^2+^ and Mg^2+^ ions [96]. Also, a decrease in the ATP/ADP ratio in the mitochondrial matrix due to increased energy consumption and a concomitant decrease in membrane potential in the mitochondria of activated neurons also increases the GDF content for substrate phosphorylation, which will also contribute to increased KGDHC activity.

The literature presents the results of animal and human studies showing the antioxidant properties of AKG; namely, a number of metabolites modify the activity of KGDHC, including inactivation by 4-hydroxynonenal and other reactive oxygen species (ROS) [72]. The mechanism of action of AKG is associated with the metabolism of amino acids, such as glutamate and glutamine, and has nutritional and therapeutic effects via the glutamine-AKG axis, improving the health and well-being of animals and humans [97]. AKG acts as an antioxidant because it reacts directly with hydrogen peroxide to form succinate, water, and carbon dioxide. The mechanisms of oxidative decarboxylation involving AKG are also considered as important [98].

Another study also discusses improvement of cellular energy status, immunity, and health via animal and human nutrition [99]. The possible physiological mechanism of the effect of AKG on the digestive tract was elucidated by scientists in a study on Cherry Valley ducks in the following way: the ratio of AMP to ATP, total adenine nucleotide in the ileal mucosa and hepatic and ileal messenger RNA expression of AMP kinase α-1, and hypoxia-inducible factor-1α [100].

The effects of AKG on ageing processes and healthspan and the possibilities of AKG use as an anti-ageing agent are shown in Table 1.

AKG exerts its pro-apoptotic effect with an effective anti-metastatic potential ratio [111]. AKG decreased oxygen consumption and increased autophagy processes through activation of AMPK signalling and inhibition of the mTOR pathway [105]. AKG can modulate energy production mechanisms connected with TAC functioning and the production of moderate ROS levels according to the hormesis conception [106] and has antioxidant properties [103].

Scientists discuss the involvement of AKG in multiple metabolic and cellular pathways as an important factor in amino acid biosynthesis, epigenetic processes, cellular signalling, transcription in cancer development and progression, protein deficiency oxidative stress conditions, and as an immunomodulatory agent [108,109], which was shown in roundworm *Caenorhabditis elegans*, *Drosophila*, mice, and humans used as models in different ageing and longevity studies [104].

## 8. Succinic Acid and Age-Related Pathologies

Physiological hypoxia and low oxygen levels lead to reduced activity of the SDH enzyme, which metabolises SC, and other oxygen-dependent enzymes in the electron transport chain, causing SC accumulation [112]. There are many data on the production of ROS by intact mitochondria in different tissues (skeletal muscle, heart, and liver of rats) depending on the goals of the experiment and the use of various substrates and inhibitors of the electron transport chain. It has been shown that mitochondria do not release measurable amounts of superoxide or hydrogen peroxide when respiring on complex I or complex II substrates. Importantly, skeletal muscle or cardiac mitochondria generated significant amounts of superoxide from complex I through palmitoylcarnitine as substrate respiration. The authors concluded that, in physiological conditions, mitochondria do not produce significant amounts of ROS [113].

Another analysis [114] of the rate of superoxide/H_2_O_2_ production from different sites of rat skeletal muscle mitochondria oxidising substrates showed that, when succinate was oxidised, most of the superoxide formation came from the quinone reduction site in complex I (site IQ), but when glutamate and malate were used as substrates, the IQ site made little or no contribution. This supports conclusions about the preferential mobilisation of the respiratory chain by succinate (experiments without inhibitors) compared to alpha-ketoglutarate for the production of AFC in muscle tissue [114].

Some studies have demonstrated the specific role of SC in cancer development associated with the discovery of pseudohypoxia phenomena [115], which refers to the activation of hypoxia signalling pathways under normal oxygen levels [116]. Pseudohypoxia is a typical event in tumours with mutated SDH and has been shown in many studies [117]. Mutations in complex II-succinate dehydrogenase, a tumour suppressor, were shown to stabilise HIF-1 and the related pseudo-hypoxia condition, and on the other hand, to prevent pseudo-hypoxic gene expression in aerobic cardiac cells [118].

The literature collects the results of animal and human studies that indicate that targeting metabolic dysregulation has significant implications for the treatment of age-related cardiac fibrosis and diastolic dysfunction by the oxidation and concentration of SC in the heart of old animals. A novel mechanism by which succinate induces fibroblast activation and apoptosis resistance by promoting PKM2 dimerisation in the heart has been demonstrated [88]. Hence, the inhibition of SC generation or blocking its downstream effects is potentially a promising new strategy for slowing down heart ageing and kidney ischaemia–reperfusion injury [89].

The effects of the metabolic syndrome associated with increased glucose levels, which are relieved by metformin, can be considered as disturbances in energy metabolism and glucose utilisation through insulin resistance, which is a frequent associated factor of age-related changes [119]. These effects can be achieved through the application of metformin, which is recognised as a potential anti-ageing agent [120]. It is not important that the induced intense oxidation of succinate in mitochondria under hypoxic loads, which often accompany pathological conditions, is accompanied by a significant increase in the production of free radicals and mitochondrial dysregulation processes [89]. The succinate-dependent metabolism pathways during ageing and pathologies with a varied genesis are shown in Table 2.

According to one of the postulates of the ageing theory, chronic inflammatory processes, especially severe inflammation activating the hypothalamic–pituitary–adrenal axis, are accompanied by the production of anti-inflammatory glucocorticoids by the adrenal cortex, including suppression of the TCA cycle and oxidative phosphorylation in mice [128]. These changes in adrenal dysfunction during severe inflammation at the level of SDH lead to suppression of ATP synthesis and SC accumulation and are accompanied by activation of enhanced ROS generation. Thus, an effective therapeutic way to eliminate anti-inflammatory dysfunctions of the adrenal gland at increased levels of SC, which disrupts oxidative phosphorylation and ATP synthesis, is to reduce the SC level.

## 9. Different Levels of Initial Physiological Reactivity of the Organism Potentially Determine the Mechanisms of Ageing

It has been shown that high and low physiological reactivity of organism systems to the action of different adverse factors (stress, adaptation, resistance to hypoxia, etc.) depends on the intensity of oxygen consumption, metabolic peculiarities, behavioural reactions, and a number of other individual differences [16,17,18,20]. A long-term study of monozygotic twins conducted for 8 years at sea level and altitude has shown that the respiratory response to a hypoxic stimulus is a rigid, genetically determined, physiological characteristic reflecting the general non-specific reactivity of the organism [122]. In particular, it has been shown that the human ability to maintain relatively constant levels of oxygen consumption under hypoxia is genetically determined (70–80%) and depends on individual sensitivity to hypoxia and hypercapnia. The genetic determinacy of hypoxia tolerance is confirmed by data showing numerous polymorphisms of the HIF1A gene [129]. These features of energy metabolism and organism responses are best manifested in age-related changes, thus inducing individual programmes of age-related changes. However, genetic factors are one of the leading factors, but not less important are the traits of each individual that actively counteract these unfavourable age-related changes—hence, the importance of social programmes for the elderly implemented in many countries. This is important now, given the increasing ageing of the population in many countries [1,2,7,62].

Sex and age differences in the ability to adapt to high altitudes have been revealed, when it was shown that females adapted more quickly and more easily to acute mountain sickness compared to males [130], and highly hypoxia-resistant individuals predominated among female rats, while males turned out to be predominantly low-resistant and medium-resistant [131]. These factors have been shown to be mechanisms of distribution of different mortality rates among older males and females. There is no doubt that resistance to the action of hypoxic factors with a varied genesis is one of the leading mechanisms in the initiation of pathological conditions (metabolic acidosis, increased tension of the oxygen transport system, hyperlipidemia, and reduced activity of antioxidant systems in counteracting oxidative stress).

Also, these ageing mechanisms are connected with different levels of sensitivity of internal organs to hypoxia and, accordingly, ageing, as bones, skeletal muscles, thymus, and spleen exhibit high resistance to hypoxia, whereas brain, heart, kidneys, lungs, and liver are characterised by low resistance [16,17]. Probably, the range of these highly affected organism systems in ageing is more pronounced, when taking into account the initial level of metabolism under the influence of unfavourable factors, such as emotional status, especially in the conditions of intensification of modern life and a high-fat diet, and the influence of anthropogenic factors of different origins connected with the globalisation of economic interrelations.

The available studies predominantly consider only two extreme variants of this situation—high and low reactivity of physiological systems. However, a large group of moderately resistant individuals have both groups of traits. The use (or ignorance) of effective programmes to counteract these stresses (measures related to the prevention of ageing, such as rational nutrition with calorie restriction, physical and mental exercise, fitness, group therapies, and many other self-organisation techniques, etc.) leads to the transition from one extreme group to the other only through the group of moderately resistant individuals. This is important, because the characteristics of animals with low resistance to hypoxia, established in experiments, have shown them as individuals with a weak nervous system, increased emotional reactivity, less developed internal inhibition, increased excitability, rapid exhaustion of the excitatory process, and a high predisposition to the development of such diseases as diabetes, obesity, thyrotoxicosis, atherosclerosis, etc. [17]. Thus, such conditions of physiological functioning of organisms in high-risk groups are certainly associated with decreased survival and limited years of active ageing.

## 10. Individual Ageing, Energy Metabolism, Hormonal Status, and Receptor Control

The interaction between energy metabolism, hormonal status, and the receptor system of the cell can be affected by pathological factors, as convincingly demonstrated in studies of hypoxia of varied genesis [19,132,133]. It is known that hypoxic damage, stress, and other conditions, especially in the course of ageing, are closely related. Accordingly, a positive relationship between the predominant oxidation of SC in TAC and catecholamines has been established, and, conversely, exogenous SC can stimulate catecholamine (CA) metabolism under stressful loads. The existence of such bilateral relationships suggests the direct involvement of SC in the processes of regulating synaptic transmission. In turn, a number of the effects of this system are reciprocated by another reaction, i.e., the specific activation of AKG oxidation by acetylcholine (ACh) in the Mt. This system, opposite to succinate, is driven by the activation of aminotransferase reactions, while inhibiting the activity of SDH under functional loads of varied nature (hypoxia, stress, etc.). On the other hand, exogenous AKG has cholinomimetic properties, exerting an effect on the ACh-cholinesterase system [134,135,136]. These two multidirectional systems are important for determining individual adaptation to low-oxygen conditions in the environment, which has been convincingly demonstrated in animals and humans with different levels of initial resistance to hypoxic and other loading [16,17,18,20,133,136].

This high individual reactivity is associated with the functioning of the ACh-cholinesterase system, which activates the part of oxidative processes that allows the use of the nitrate–nitrite component of cellular reactions under hypoxia. This approach is essential for improvement of the survival of this group of animals under acute hypoxia. In low-resistant animals, the oxidation of succinate under acute oxygen deficiency is dominated mainly by the intensive metabolism of catecholamines, accompanied by critical changes in the oxygen energy supply. In the group of highly resistant animals, the predominant oxidation of alpha-ketoglutarate and activation of the ACh system result in the induction of the nitrite–nitrate respiration component [137].

This important element becomes crucial for maintaining the functional state of cholinergic receptors and determines metabolic cellular and mitochondrial rearrangements in conditions of acute oxygen deficiency not only for ACh but also for NO, the volume of which increases significantly during adaptation to hypoxia. Precisely these mechanisms of the formation of effective nitric oxide depots provide effective preconditions for increasing metabolic reserves during the physical training of older people and prevent the onset of cardiovascular pathologies. The relationship between nitric oxide and vascular pathologies in older adults has been convincingly demonstrated in a number of studies [20,136] and is shown in Figure 3. Physical activity in older people is important for maintaining the correct physiological ageing process [10].

It is important to note the connection between theories of ageing and both free radical and mitochondrial dysfunction, which has been the subject of many studies [31,33]. Thus, the treatment with the main intermediates of the Krebs cycle, which, in turn, can modify NO production [137], in order to correct the energy supply, first of all causes changes in the functional state of the Mt themselves, which can switch to reducing energy consumption and, at the same time, to reducing the production of ROS [72,106]. Reduced ROS production is an important factor in the protective effect of Krebs cycle substrates, as well as NO donors, in order to utilise oxygen economically. Therefore, the problem of increasing the efficiency of mitochondrial oxidative phosphorylation and energy supply processes during the ageing period and maintaining the physiological ageing process is associated with the interdependence of these processes, when changes in the functional state of the Mt act as factors in the production of ROS, and the latter are also able to modify the energy supply processes by the main TAC intermediates (Figure 4).

The nature of the action of various physiologically active substances, in particular TAC intermediates SC and AKG, and other compounds that lead to the formation of these substances, in aminotransferase and other reverse reactions as a promising means to prevent damage to various body systems during ageing has attracted the attention of many researchers [88,89,101,102,103,105,121]. First of all, this concerns the use of SC to normalise various pathological abnormalities associated with ageing in clinical and animal studies [112]. The effect of other TAC substrates, namely AKG, on changes in the functional state of the most important body systems has also been studied [109]. In particular, the introduction of AKG provided for the efficiency of the cardiovascular system prevented the appearance of stress-related tissue damage [19] by redistributing metabolites in functional tissues through the activation of the cholinergic mechanism of body regulation [135]. In these conditions, the transaminase pathway of TAC substrate supply was activated, which increased the energy supply in cells [138].

## 11. Biomarkers of Ageing and the Krebs Cycle

With age, human cells react to stress more actively, become overworked, and the ageing process accelerates. What is the role of Krebs cycle intermediates in these stress-preventing processes that accelerate ageing and prevent active longevity? The theories of ageing are based on comprehensive genomic screening studies that have shown that the ageing process is associated with significant epigenetic changes in the chromatin landscape, such as global demethylation of DNA and histones and increased histone acetylation [139,140].

These mechanisms of control of gene expression by the modification of the epigenetic landscape of chromatin are important as regulatory mechanisms of the key mechanisms of energy supply induced by the Krebs cycle. From this point of view, studies on the effects of TAC intermediates, such as α-ketoglutarate, succinate, and fumarate, which can regulate DNA and histone methylation levels, and citrate, which can also enhance histone acetylation, are important. This relationship is seen through the effects of DNA demethylase (TET1-3) and histone lysines (KDM2-7), which are members of 2-oxoglutarate-dependent dioxygenases (2-OGDO). The 2-OGDO enzymes are activated by oxygen, iron, and α -oxoglutarate, and these in turn are inhibited by succinate and fumarate [141,142].

The dependencies of mitochondrial energy supply processes shown in the literature demonstrate important differences in the use of Krebs cycle intermediates for the formation of adaptation reactions of oxygen supply [16,17], which can change significantly during ageing and have been confirmed in experiments on the accumulation and efficient removal of succinate in many pathologies [88,89]. Thus, time-critical compensatory reactions at the early stage of hypoxia are realised in the conditions of suppression of NADH-oxidase oxidation through the activation of the succinate oxidase oxidation pathway. The latter is also necessary for the formation of regulatory bioenergetic mechanisms underlying long-term adaptation during the transition period. This refers to the period of quantitative and qualitative changes in the properties of respiratory chain enzymes and the interaction of mitochondrial enzyme complex I and II aimed at restoring the NAD-dependent oxidation pathway. The completion of the formation of bioenergetic mechanisms of long-term adaptation is associated with the restoration of the NAD-dependent oxidation pathway and the loss of importance of succinate oxidase oxidation [16,17]. Therefore, the short-term and long-term bioenergetic mechanisms of adaptation to hypoxia implemented during the life of each individual differ significantly, which in turn can intensify the ageing programme.

Thus, Krebs cycle metabolites may be involved in age-related abnormalities of the mitochondrial metabolism and, in this way, may induce epigenetic reprogramming that contributes to the senile phenotype and degenerative diseases. The metabolism of these compounds is particularly important when considering the variety of neurological diseases observed in old age, including stroke, traumatic brain injury, and Alzheimer’s disease, which may be associated with adaptive gene expression to protect the nervous system in these conditions [143].

## 12. Neurohumoral Regulation, Metabolic Disorders, Substrates of the Krebs Cycle

Recently, in the care of patients with cardiac pathology, increasing attention has been focused on the use of metabolic agents that include Krebs cycle substrate derivatives [144]. These succinate-based drugs, as convincingly shown in a number of studies [145], do not affect the causative factors of coronary heart disease, but significantly normalise metabolic disorders directly related to its pathogenesis [146]. Equally important, they also target the concomitant hyperactivity of neurohumoral regulation mechanisms.

It is known that, in the conditions of normal blood supply to the heart, the main source of its energy supply is the utilisation of free fatty acids (FFA), especially long-chain fatty acid derivatives shown as myocardial metabolic imaging agents [147], which gives the maximum energy yield per unit of substrate and provides 60–90% of ATP necessary to maintain myocardial function and life support. The presence of FFAs as a substrate of energy metabolism is accompanied by a sharp inhibition of glucose and lactate oxidation in the myocardium [148]. Therefore, a decrease in the concentration of FFAs in the blood plasma, or pharmacological effects that directly inhibit their oxidation in the mitochondria, as shown for malonyl CoA decarboxylase, protects the ischaemic heart and leads to an increase in the rate of mitochondrial pyruvate transport and, as a result, an increase in glucose and lactate utilisation [149]. Important issues related to pyruvate metabolism as a key pathway for glycolysis and oxidative phosphorylation, which is crucial for energy homeostasis and mitochondrial quality control, are reviewed in study [150]. Pyruvate can induce the accumulation of ROS and induce the flux of calcium ions into the mitochondria. These effects of pyruvate can result in mitochondrial ultrastructural changes, mitochondrial dysfunction, and metabolic dysregulation. An important role of pyruvate is related to the processes of the fusion dynamics, fission, and mitophagy of these cellular structures [150].

At the same time, the oxidation of FFAs is associated with the consumption of large oxygen amounts, and the limitation of its supply in coronary vascular disease is accompanied by a sharp impairment of FFA utilisation as a substrate of energy metabolism in cardiomyocytes [151,152]. Then, oxidised metabolic products accumulate in cardiomyocytes, inhibiting mitochondrial translocation and the transfer of macroergic phosphates across their membrane. This is also accompanied by a decoupling of the reaction of oxidation and phosphorylation and causes a sharp energy deficit [153]. As a result, aerobic glucose utilisation is also blocked, and myocardial energy supply results in an inefficient anaerobic pattern. Such changes in the heart tissue are associated with the activation of glycolysis, the accumulation of lactate and protons in the cytosol, the development of acidosis, and a subsequent inhibition of cardiomyocyte contractile function. Therefore, one of the possible mechanisms for maintaining myocardial energy metabolism in the presence of oxygen deficiency may be blocking the utilisation of FFAs, which allows more complete aerobic glucose utilisation, which is possible even with a limited oxygen supply. One of the pharmacological ways to solve this problem is the use of SC, which has previously been shown to have a pronounced antihypoxic [154] and antioxidant effect, especially on ischaemia–reperfusion injury [123,155].

It has been noted that SC activates the SDH pathway of glucose oxidation, which switches the cellular metabolism under hypoxia to a more oxygen-saving direction of energy metabolism [156]. In addition, as an effective metabolic substrate, SC improves energy metabolism by optimising the functioning of the mitochondrial respiratory chain, which helps to stabilise the cell membrane and reduce post-hypoxic metabolic acidosis. Succinic acid has also been shown to increase ATP synthesis, inhibit glycolysis, activate aerobic processes in cells, enhance gluconeogenesis, and stabilise cell membranes [157,158]. The antioxidant effect of SC is related to its ability to bind free radicals, inhibit free radical processes, and increase the activity of antioxidant enzymes. This leads to a wide range of applications for succinate and other derivatives of TAC [159] in cardiological and other treatments, especially in older patients with severe cardiac pathologies [160].

## 13. Conclusions

Summarising, it can be concluded that anti-ageing biology and medicine programmes define ageing in different categories and explain the interactions between various factors stimulating the development of age-related changes and diseases. This paper presents several theories of ageing, with the major causes of the process related to epigenetic changes associated with gene expression, global DNA methylation, epigenetic alterations, genomic instability, mitochondrial dysfunction, depletion of telomeres on chromosomes or telomere attrition, changes in proteostasis processes, uncontrolled free radical production, and oxidative stress. This review presents clinical descriptions of the major markers of aging currently used to screen for DNA copy number, telomere length, global DNA methyl thione: Hsp70 and Hsp72, insulin-like growth factor 1 and SIRT1 (Sir2) (i.e., NAD+-dependent deacetylase that plays a critical role in a wide range of biological events), deregulated levels of nutrient sensitivity, circulating osteogenic progenitor cells as an indicator of stem cell depletion, and IGF-1, SIRT1, GDF-15, IL-6, CRP and TNF-alpha. Additionally, more evidence about these biomarkers is presented.

This review contains data concerning the analysis of inflammatory processes that accelerate ageing and levels of glycosylated haemoglobin, IL-6, C-reactive protein, ferritin, and homocysteine. Accelerated ageing and age-related pathology were discussed with reference to important Krebs cycle metabolites (α-ketoglutarate (AKG) and succinate (SC)) and their role in energy metabolism and ageing. This has been demonstrated in AKG and SC studies on different species (roundworm *Caenorhabditis elegans*, *Drosophila*, mice, and humans used as models).

This review describes convincingly the differences in the mitochondrial characteristics of energy metabolism in animals with different levels (high and low) of physiological reactivity of functional systems related to the state of different regulatory systems providing oxygen-dependent processes. Much attention is given to the crucial role of tricarboxylic acid (TCA) cycle intermediates, i.e., α-ketoglutarate (AKG) and succinate (SC), in the energy metabolism in cells related to amino acid and protein synthesis, epigenetic regulation, cell stemness and differentiation, as well as metabolism associated with the development of pathological conditions and, in particular, cancer cells. This article also addressed the issue of ageing in terms of individual characteristics related to physiological reactivity. This review has demonstrated the role of the Krebs cycle as the key component of cellular energy and ageing, which is closely associated with the development of various age-related pathologies, such as cancer, type 2 diabetes, and cardiovascular and neurodegenerative diseases where the mTOR pathway plays a key role in the regulation of ageing.

The findings suggest that this approach may also be useful to show that Krebs cycle metabolites may be involved in age-related abnormalities of the mitochondrial metabolism and may thus induce epigenetic reprogramming that contributes to the senile phenotype and degenerative diseases. The metabolism of these compounds is particularly important when considering ageing mechanisms connected with different levels of initial physiological reactivity and able to initiate individual programmed ageing depending on the intensity of oxygen consumption, metabolic peculiarities, and behavioural reactions.

## Figures and Tables

**Figure 1 biomolecules-14-00260-f001:**
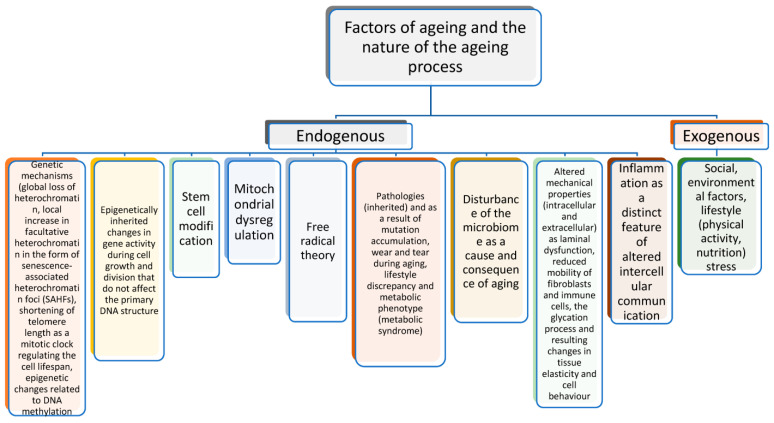
Biological mechanisms of the ageing process.

**Figure 2 biomolecules-14-00260-f002:**
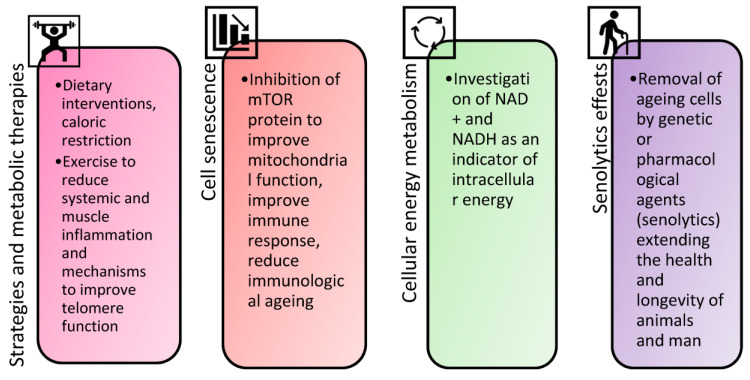
The main areas of research regarding lifespan and the continuation of a healthy life.

**Figure 3 biomolecules-14-00260-f003:**
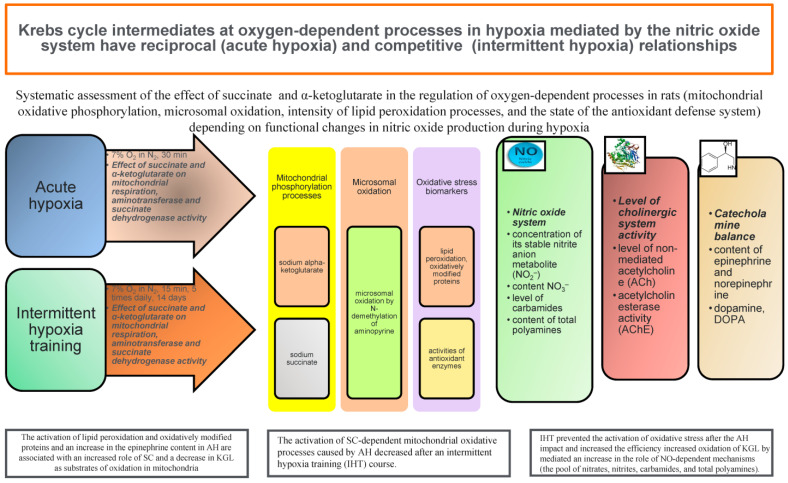
Systematic assessment of the effect of succinate and α-ketoglutarate in the regulation of oxygen-dependent processes in rats depending on functional changes in nitric oxide production during hypoxia [19].

**Figure 4 biomolecules-14-00260-f004:**
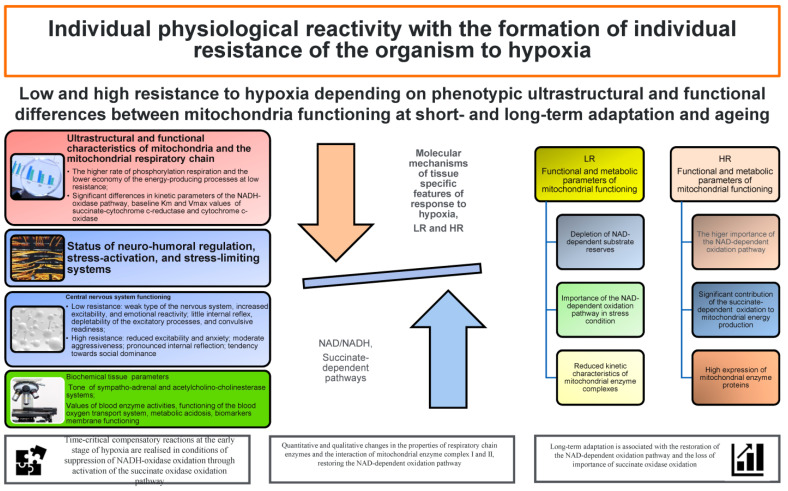
Low and high resistance to hypoxia by phenotypic ultrastructural and functional differences between mitochondria functioning in short- and long-term adaptation and ageing [16,17,18,19].

**Table 1 biomolecules-14-00260-t001:** Effects of alpha-ketoglutarate (AKG) on ageing processes and healthspan and possibilities of AKG use as an anti-ageing agent.

Model	Description	Possible Mechanisms	Authors
Lifespan of adult *Caenorhabditis elegans*, proposal of new strategies for the prevention and treatment of ageing and age-related diseases in both *C. elegans* and mammalian cells	Molecular mechanisms underlying ad libitum feeding processes, dietary restriction connected with the lifespan and age-related diseases in evolutionarily diverse organisms	AKG impact inhibited ATP synthase and reduced the ATP content, which decreased oxygen consumption and increased autophagy processes	[101]
*Caenorhabditis elegans* as a model system, *C. elegans* orthologue of the ATP5B model, effects of modulators of worm longevity	Critical role of AKG in prolonging the lifespan, mediator of autophagy by inhibition of the TOR mechanism	ATP synthase as a potential target, mechanism of inhibition of TOR by AKG, the relationship of ATP58 inhibition with TOR target of rapamycin (TOR), 5′ adenosine monophosphate-activated protein kinase (AMPK), and FoxO	[102]
*Drosophila* fruit fly model receiving diets with 5 μM AKG	Heat shock protein genes (Hsp22 and Hsp70), mRNA expression, cry, FoxO, HNF4, p300, Sirt1, AMPKα, HDAC4, PI3K, TORC, PGC, and SREBP genes	ATP/ADP ratio, increased autophagy processes, AMPK, activation of AMPK signalling, inhibition of the mTOR pathway	[103]
Analysis of AKG effects on roundworm *Caenorhabditis elegans* (*C. elegans*), *Drosophila*, mice, and humans as models in different ageing and longevity studies	The review discusses different AKG effects on the lifespan depending on the animal (*C. elegans*, *Drosophila*, mice) and human models	Regulation of stem cell behaviours, modulation of the level of inflammation processes, activation of the Nrf2/ARE signalling pathway, and ROS reduction, the AMPK signalling pathway and downregulation of mTOR	[104]
Potential therapeutic use in humans to treat age-related diseases, clinical studies of therapeutic interventions	Potential positive effects of AKG on muscle growth, wound healing, and promotion of faster recovery after surgery; dietary supplementation in humans	Antioxidant properties; cellular respiration processes, one of the key regulators of the citric acid cycle	[105]
Analysis and discussion of data from different sources obtained in different models: nematodes, fruit flies, yeast, and mice, and limitations of AKG use as a geroprotective agent	Potential anti-ageing effects and geroprotective action of AKG through mimicking of calorie restriction and properties of a hormesis agent	Modulated energy production mechanisms connected with Krebs cycle functioning, production of moderate levels of ROS according to the hormesis conception, impact on DNA obligate substrate and histone demethylases processes, direct antioxidant properties	[106]
Analysis of alpha-ketoglutarate calcium salt on the healthspan and lifespan in C57BL/6 mice	Series of longitudinal and clinical experiments on a longer and healthier life in the murine model, by a mechanism reducing frailty and enhancing longevity, and compression of morbidity	Decrease in the levels of systemic inflammatory cytokines, namely IL-10	[107]
Calcium alpha-ketoglutarate salt dietary supplementation in the murine model, frailty index study, analysis of the efficacy of processes in boosting health and longevity models	Significant increase in the median and maximal lifespan in mice, a decrease in the proportion of life in which mice were frail, reduction in frailty scores of females and males after the impact of the AKG diet. Only AKG-fed females were protected against age-dependent increases in circulating inflammatory cytokines	AKG as an important agent of the ageing regulatory pathway, an amino acid metabolism player, and a partner in aminotransferase reactions, reactions involved in chromatin modification, immune and inflammatory pathways, growth regulation, and epigenetic regulation of gene expression	[108]
Therapeutic use of AKG in different metabolic pathological processes and for treatment of diseases	The involvement of AKG in multiple metabolic and cellular pathways; the metabolite as an important key factor in amino acid biosynthesis, epigenetic processes, cellular signalling, a transcription factor in cancer development and progression, protein deficiency oxidative stress conditions, an immunomodulatory agent, and a bone anabolic factor	Endogenous intermediary metabolite in the Krebs cycle, hydroxylation reactions on various types of substrates, hypoxia-inducible factor, oxidative stress	[109]
Clinical study testing in vitro methods; 28 days of treatment with the use of AKG-containing cream; epidermal keratinocyte proliferation assays	Assessment of skin wrinkles, texture, elasticity, and firmness in vitro using capillary electrophoresis time-of-flight mass spectrometry assays and rice-fermented liquid	AKG significantly reduced skin wrinkles and had anti-ageing effects on epidermal keratinocyte proliferation in the skin	[110]
Human osteosarcoma (OS) cell lines Saos-2 (HTB-85TM) and HOS (CRL-1543TM) model; alpha-ketoglutarate disodium salt dihydrate impact, in vitro study	Anti-osteosarcoma effects of AKG supplementation in an in vitro study analysis; JNK pathway, Bax/Bcl-2 ratio, caspase-9 and caspase-3; pro-metastatic TGF-β, and pro-angiogenic VEGF cytokines	Pro-apoptotic effect of AKG, with its anti-metastatic potential linked with inhibition of OS cell motility	[111]

**Table 2 biomolecules-14-00260-t002:** Succinate-dependent metabolism pathways during ageing and pathologies with varied genesis.

Model	Description	Possible Mechanisms	Authors
SUCNR1^−/−^ mouse model and an AAV9-based approach, older mice/human model, succinate receptor as SUCNR1/GPR91 in analysis of fibrosis processes in old animals, diastolic dysfunction process depending on age	Succinate promoted fibroblast activation and apoptosis resistance in both young and old mice via succinate receptor SUCNR1 and stimulation of PKM2 dimerisation	Dimeric PKM2 translocated to the nucleus and mitochondria, where it promoted fibroblast activation and apoptosis resistance via interaction with HIF-1α; the metformin impact as a mediated succinate-dependent mechanism agent may inhibit fibroblast activation and apoptosis resistance in a murine model	[88]
In vitro model of ischaemia–reperfusion kidney injury in mice; proximal tubule cell-specific Pdk4 knockout (Pdk4^ptKO^) murine model, pyruvate dehydrogenase kinase 4 deficiency analysis	Knockout or pharmacological inhibition of the PDK4 pathway ameliorated ischaemia–reperfusion kidney damage caused by a cell-permeable form of succinate, i.e., dimethyl succinate, and mitochondrial ROS generation processes	Inhibition of PDK4 prevents in vitro ischaemia–reperfusion kidney injury via the reduction in succinate accumulation and mitochondrial dysfunction	[89]
Analysis of postoperative cognitive dysfunction processes, gerontological patients after cardiac surgery, and a cognitive impairment model, people over 60 years of age, Cytoflavin containing succinic acid	Cytoflavin containing inosine, nicotinamide, riboflavin, and succinic acid was used in elderly postoperative patients in a multicenter, double-blind, placebo-controlled, and randomised study	Improvement of gerontological patients’ condition	[121]
Study of steroidogenic adrenocortical cells in LPS-induced systemic inflammation processes in a murine model, succinate–succinate dehydrogenase relationship	Increased succinate levels by disruption of oxidative phosphorylation and increased ATP synthesis connected with high ROS production	SDHB expression via upregulation of DNA methyltransferase 1 and methylation processes in the *SDHB* promoter	[122]
Review analysis of the physiological and pathophysiological condition connected with succinic acid metabolism and SDH functions	Succinate functions and hypoxia-inducible factor (HIF)-1α, development of pseudohypoxia and tumours via mutated SDH, succinate functions in metabolic or non-metabolic pathways, lysine succinylation process as proteins and immunomodulatory modification levels, blood formation or haematopoiesis	Activation of succinate receptor 1 (SUCNR1), G protein-coupled receptor 91 (GPR91), or hypoxia-inducible factor-1α, (HIF)-1α	[112]
In vivo ischaemia–reperfusion of heart in an open-chest mouse model, metabolomic analysis of ex vivo Langendorff heart experiments	Study of succinate-dependent mitochondrial superoxide production in myoblasts	Inhibition of ischaemic succinate accumulation and its oxidation as an effective way in ischaemia–reperfusion conditions	[123]
Immune-defective ageing murine model, clinically relevant BRAF^V600E^ mutated YUMM1.1 melanoma tumour model, cancer immunotherapies	Tumour microenvironment study using polyethylene succinate microparticle biomaterial	Succinate-mediated immune and cancer cell responses in a tumour model and immunotherapies	[124]
Hippocampus of different aged APP/PS1 double transgenic AD mice, analysis of the β-amyloid level with the immunohistochemistry method	3, 6, 9, and 12-month-old mice groups, learning and memory test analysis, mitochondrial damage, and autophagosome accumulation assays	Abnormal accumulation of succinic acid and citric acid associated with age-related damage to hippocampal mitochondria in the APP/PS1 murine model	[125]
Comparative approach aimed at determination of the plasma methionine metabolic profile using an LC-MS/MS platform from 11 mammalian species with a longevity ranging from 3.5 to 120 years	Species longevity-specific plasma profile of methionine metabolism dependencies	Longevity connected with reduced succinate and malate levels	[126]
Mature female *Xenopus laevis* frogs INDY-expressing oocyte model, longevity gene Indy	Succinate-stimulated [14C] citrate efflux	Longevity gene Indy functions as an exchanger of dicarboxylate and tricarboxylate Krebs cycle intermediates	[127]

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
