# Peer review of "Tricarboxylic Acid Cycle Intermediates and Individual Ageing"

_biomolecules, 2024, doi:10.3390/biom14030260_

Round 1

Reviewer 1 Report

Comments and Suggestions for Authors

Overall, this is an informative article with some interesting and novel ageing perspectives. However, there are some issues with the manuscript that need to be addressed before it is in a suable condition for publication.

 Major issues

Introduction

In the first paragraph of the introduction, I believe that it is very important to have one or two sentences which mention negative senescence/negligible senescence.  Moreover what about organisms that are suggested to be immortal?

2 Theories of Ageing

I disagree that this is a theory of ageing. It is a description of a collection of biological mechanisms which have been associated with the ageing process. If it is a theory, how does it account for the evolutionary origins of the ageing process. At best it is a theory as to how the ageing process unfolds but it is dose not explain why humans and other organisms for that matter age.  

Following on from the above point. There is a lack of recognition of ageing from an evolutionary perspective in this section.  I very much think that it should be at least briefly mentioned. Recent reviews which discuss the evolution of ageing below may help you do this.

https://www.sciencedirect.com/science/article/pii/S156816372100235X

https://royalsocietypublishing.org/doi/10.1098/rspb.2019.1604

https://www.sciencedirect.com/journal/experimental-gerontology/special-issue/109WWQFWVD2

In line 164 you state

“The main causes of ageing today include…..”

This is quite an authoritative statement to make. Arguably much about the ageing process remains to be determined. Yes, there are certainly mechanisms and processes associated with ageing but to definitively state that there is a cause for ageing is probably a little bit too strong of a statement to make.

Line 213 this is quite a controversial statement to make. As above it is a very authoritative statement.

The term "anti-aging" has firmly entered our lives, and nowadays there is no need to

explain to the patient what exactly an anti-ageing specialist does

section 9

citations are needed here. These statements really need backed up

 “It has been shown that high and low physiological reactivity of organism systems to 453 the action of different adverse factors (stress, adaptation, resistance to hypoxia, etc.) de- 454 pends on the intensity of oxygen consumption, metabolic peculiarities, behavioural reac- 455 tions, and a number of other individual differences” 

 Minor issues

Introduction

Citation(s)

Required to backup this statement “have been convincingly shown in the literature to prolong longevity and/or health.”

Please do not use abbreviation in the highlights. Use the full name. This will make the manuscript easier to read.

Figure 2 please change “the lifespan” to “lifespan”

  Please consider making the abstract more concise. At present it is somewhat long and as I have stated above this will improve the accessibility and readability of your manuscript

Comments on the Quality of English Language

Overall, this is an informative article with some interesting and novel ageing perspectives. However, there are some issues with the manuscript that need to be addressed before it is in a suable condition for publication.

 Major issues

Introduction

In the first paragraph of the introduction, I believe that it is very important to have one or two sentences which mention negative senescence/negligible senescence.  Moreover what about organisms that are suggested to be immortal?

2 Theories of Ageing

I disagree that this is a theory of ageing. It is a description of a collection of biological mechanisms which have been associated with the ageing process. If it is a theory, how does it account for the evolutionary origins of the ageing process. At best it is a theory as to how the ageing process unfolds but it is dose not explain why humans and other organisms for that matter age.  

Following on from the above point. There is a lack of recognition of ageing from an evolutionary perspective in this section.  I very much think that it should be at least briefly mentioned. Recent reviews which discuss the evolution of ageing below may help you do this.

https://www.sciencedirect.com/science/article/pii/S156816372100235X

https://royalsocietypublishing.org/doi/10.1098/rspb.2019.1604

https://www.sciencedirect.com/journal/experimental-gerontology/special-issue/109WWQFWVD2

In line 164 you state

“The main causes of ageing today include…..”

This is quite an authoritative statement to make. Arguably much about the ageing process remains to be determined. Yes, there are certainly mechanisms and processes associated with ageing but to definitively state that there is a cause for ageing is probably a little bit too strong of a statement to make.

Line 213 this is quite a controversial statement to make. As above it is a very authoritative statement.

The term "anti-aging" has firmly entered our lives, and nowadays there is no need to

explain to the patient what exactly an anti-ageing specialist does

section 9

citations are needed here. These statements really need backed up

 “It has been shown that high and low physiological reactivity of organism systems to 453 the action of different adverse factors (stress, adaptation, resistance to hypoxia, etc.) de- 454 pends on the intensity of oxygen consumption, metabolic peculiarities, behavioural reac- 455 tions, and a number of other individual differences” 

 Minor issues

Introduction

Citation(s)

Required to backup this statement “have been convincingly shown in the literature to prolong longevity and/or health.”

Please do not use abbreviation in the highlights. Use the full name. This will make the manuscript easier to read.

Figure 2 please change “the lifespan” to “lifespan”

  Please consider making the abstract more concise. At present it is somewhat long and as I have stated above this will improve the accessibility and readability of your manuscript

Author Response

Dear Editor,
I thank you and the anonymous reviewers for your excellent feedback to my manuscript and for your constructive comments.

These comments are very important for improving the quality of my manuscript and for my follow-up studies.

I have extensively revised my manuscript based on your important comments.

Thanks.

Reviewer 2 Report

Comments and Suggestions for Authors

In this manuscript titled "Metabolic Pharmacoremediation for Prevention of Accelerated Ageing and Maintenance of Active Longevity Using Krebs Cycle Intermediates: Perspectives and Challenges," the author thoroughly explores the realm of anti-aging biology and medicine, integrating insights from genetics, molecular biology, and endocrinology. This review presents diverse theories and factors shaping age-related changes and diseases, encompassing epigenetic alterations, mitochondrial dysfunction, and oxidative stress. Specifically, the author descripts key markers of aging, such as DNA copy number and telomere length, global DNA methylation, Hsp70 and Hsp72, insulin-like growth factor 1, and SIRT1. Additionally, emphasizing the pivotal role of metabolic compounds like α-ketoglutarate (AKG) and succinate (SC) in cellular energy metabolism and aging, the paper underscores their relevance to diseases such as cancer and type 2 diabetes. The author then illuminates how understanding the interplay between these compounds, physiological reactivity levels, and metabolic processes can elucidate aging mechanisms. The paper proposes that Krebs cycle metabolites may contribute to age-related mitochondrial abnormalities and degenerative diseases through epigenetic reprogramming. Overall, the author underscores the significance of metabolic pathways in aging research and hints at potential therapeutic avenues to improve healthspan.

Major comments

1.    The title is too long and needs to be shortened to emphasize the main topic of this review paper.

2.    Some parts of the papers (2, 3, 4, 9, 10) are not directly related to the title of this paper and I think it will be better to be shortened substantially.

3.    As a review paper, some important papers are not cited and need to be discussed with citations. Please add missing references as follows:

a)  Add citation regarding senescence (Lee et al., 2022 PMID: 35950453, Chen et al., 2023 PMID: 37437984).

b)    Add citation regarding calorie restriction and autophagy (Uvdal  and Shashkova, 2023, PMID: 37238710; Park et al.  2022 PMID: 34949740).

c)    Add citation for the regulation of aging via insulin/IGF-1 signaling  in C. elegans (Lee et al. 2022 PMID: 36380728).

d)    Add reference for the role of pyruvate in mitochondrial quality control (Kim et al.,2023 PMID: 36756776).

Minor comments

1. Please edit word splitting in all figures. For example: In figure 1., edit word splitting for epigenetically, modification, mitochondrial dysregulation, communication, and environmental.

2. Please correct typos throughout the manuscript. Following are some examples:

1) On page 5, line 141, please add “to” after “According”.  .

2) On page 6, line 204, please correct “In” to “in.”

3) On page 10, table 1., please follow proper nomenclatures for model organisms.

4) On page 17, Figure 4., please check the word spacing. For example: coorect “functionaland” to “functional and”

3. In Figure 2. please add subtopics in vertical for the first two sections as well besides “cellular energy metabolism” and “senolytic effects”.

Comments on the Quality of English Language

NA

Author Response

Dear Editor,
I thank you and the anonymous reviewers for your excellent feedback to my manuscript and for your constructive comments.

These comments are very important for improving the quality of my manuscript and for my follow-up studies.

I have extensively revised my manuscript based on your important comments.

Thanks

Round 2

Reviewer 2 Report

Comments and Suggestions for Authors

The authors successfully addressed my concerns.